# Positional distribution of transcription factor binding sites in the human genome

Chun-Ping Yu[1]¤, Zhi Thong Soh[1], Maloyjo Joyraj Bhattacharjee[2], Wen-Hsiung Li[1,3]*

1 Biodiversity Research Center, Academia Sinica, Taipei, Taiwan, 2 Life Science Division, Institute of Advanced Study in Science and Technology, Paschim Boragaon, Guwahati, Assam, India, 3 Department of Ecology and Evolution, University of Chicago, Chicago

¤ Current Address: Research Center for Information Technology Innovation, Academia Sinica, 115 Taipei, Taiwan
* whli@uchicago.edu.tw

## Abstract

As transcription factors (TFs) play a major role in gene regulation, we studied their binding motifs (positional weight matrices, PWMs) and binding sites (TFBSs) in the human genome, and how TFs bind DNA motifs, including the involvement of binding co-factors. Using the chromatin immunoprecipitation sequencing data recently released by ENCODE (Encyclopedia of DNA Elements), we obtained new PWMs for 196 TFs and revised PWMs for 119 TFs. From these and the PWMs previously obtained for 235 TFs, we inferred the canonical PWMs for 500 TFs, including 243 new PWMs. Analysis revealed that most TFBSs are in introns (42.6%) and intergenic regions (31.6%), with only 11.3% in promoters. However, the TFBS density is considerably higher in promoters, showing a bell-shaped distribution of TFBSs with a peak at the transcription start site. Many TFBSs lie close to CTCF (CCCTC-binding factor) binding sites. Tethered binding is far more frequent than co-binding, with the latter often requiring co-factors.

## Introduction

A transcription factor (TF) regulates the expression of a target gene by binding to a DNA binding site in the gene's promoter or enhancer. A TF binding site (TFBS) is typically 5–15 base pairs (bp) long and a TF protein usually can recognize a set of similar DNA sequences, which are commonly represented by a position weight matrix (PWM) [1]. This matrix shows the nucleotide preference of the TF at each nucleotide position of the binding site. As TFBSs are essential for gene regulation, it is useful to know the TFBSs of a TF and their positional distribution in the genome [2–7]. Such data for various TFs in a genome can assist in identifying the TFs that regulate a specific gene.

The positional distribution of TFBSs has been studied in several organisms, including yeast, *Caenorhabditis elegans*, Drosophila, Arabidopsis and human [2–7], revealing that

**Data availability statement:** All relevant data are within the manuscript and its Supporting Information files.

**Funding:** National Science and Technology Council, Taiwan (NSTC 112-2311-B-001 -045)

**Competing interests:** The authors have declared that no competing interests exist.

TFBSs tend to lie in the vicinity of transcription start sites (TSSs) or in enhancer regions. In *Saccharomyces cerevisiae*, TFBSs are enriched in the region from 200 bp to 100 bp upstream (from −200 bp to −100 bp) of the TSS and has a sharp peak at −115 bp [4]. In *Arabidopsis thaliana*, the distribution of TFBSs in promoters is nearly bell-shaped with a peak at −50 bp upstream of the TSS and 86% of the TFBSs are in the region from −1,000 bp to +200 bp relative to the TSS [2–7]. In human, TFBS density is higher in the ±2kb region of the TSS with a peak at −50 bp [8]. However, data on the positional distribution of TFBSs in human are still limited, so we shall conduct a more extensive study, with special attention to their distribution in functional regions including enhancer regions, CTCF (CCCTC-binding factor) binding sites, and promoter regions.

In addition to analyzing the positional distribution of TFBSs in the genome, we shall address the issue of motif co-occurrence. Specifically, we will consider cases where the binding motif of the TF under study (TF1) and that of a TF from a different TF family (TF2) are found to co-localize within the same chromatin immunoprecipitation sequencing (ChIP-seq) peaks. (The TF families are defined according to the classification of Lambert et al. [9]) Two types of binding co-occurrence are known [1,8,10–14]: (a) co-binding, in which TF1 and TF2 bind neighboring sites in the same ChIP-seq peak region; and (b) tethered binding, in which TF1 binds to TF2 and then TF2 subsequently binds to DNA, while TF1 does not directly bind DNA [8,10]. For co-binding events, we shall study whether the interaction between the two TFs is mediated by cofactors and estimate the distance between the two motifs in the same ChIP-seq peak.

To study the above topics, a large number of known PWMs in an organism is required. Several databases are available for human PWMs [1,8,10–14]. The best quality PWMs usually come from ChIP-seq data. In our previous study we used the ChIP-seq data released by the ENCODE consortium before 2020 to infer PWMs and TFBSs for 354 TFs [8]. More recently, ENCODE has released the ChIP-seq data obtained from July 2020 to January 2023. This new set of ChIP-seq data includes the data for 385 TFs that have not been studied previously, along with new additional data for 167 previously studied TFs [8], totaling 552 TFs. Pratt et al. (2022) [13] have analyzed the data for 461 of the 552 (385 + 167) TFs, but they did not provide the canonical PWMs and did not study the positional distribution of TFBSs in the human genome. We shall analyze this set of data to infer new PWMs and also to revise the PWMs that we obtained previously using the older ENCODE data. In particular, we shall develop a rigorous procedure to identify the canonical PWMs of the TFs under study, because they are required for accurate identification of palindromes and repeats in TF binding motifs and for other purposes such as the prediction of the target genes of a TF under study. We shall then integrate the new, revised, and old PWMs to locate their TFBSs and study their positional distribution in the human genome.

## Materials and methods

### Collection of ENCODE ChIP-seq data

A total of 701 new ChIP-seq experiments on 552 TFs were released from July 2020 to January 2023 by the ENCODE consortium [15,16] (accessed January 2025). The

552 TFs included 385 TFs that were not included in the previous release and we analyzed the new data to obtain their PWMs. The 552 TFs also included new experiments for 167 TFs that were included in the previous release, and we integrated these data and the previous data to revise the PWMs we obtained previously [8]. Our previous study also included 235 TFs that have no new experiments in the recent ENCODE release; we call the PWMs for these TFs the old PWMs. We pooled the newly inferred PWMs, the revised PWMs and the old PWMs together in subsequent analyses to characterize PWMs and TFBSs.

**The computational pipeline used to infer PWMs**

We slightly revised the computational pipeline developed in our previous study [8] (https://github.com/chpngyu/chip-seq-pipeline) to analyze the collected ChIP-seq data. Briefly, the revised pipeline includes the following steps: First, the ChIP-seq reads are processed by Trimmomatic [17] and are then mapped to the human genome (GRCh38.p13) for each experimental replicate in an experiment using Bowtie2 [18]. Second, the mapped reads in the replicates of an experiment are pooled together to detect DNA-binding events (peaks) using MACS2 [19] with $q$-value<0.05. Each peak is defined by extending 100 base pairs on either side of the identified summit of a binding event. We then exclude the detected peaks in the blacklist regions [20]. Third, the sequences in the top 500 peaks (if only one experiment is available) or 1000 peaks (if two or more experiments are available) are used to find up to top 5 PWMs, each of which should be supported by 100 peaks or more, using MEME-ChIP [21]. In the above procedures, we set multiple criteria in each step for quality control, and if an experiment does not pass the criteria, we use another set of criteria to try to recover a good quality PWM --- essentially, we look for a PWM supported by >250 peaks among the peaks used (see Yu, et al., 2021 for details). If a TF has been studied in multiple experiments, often in different cell lines or tissues, we use a ranking method to select the top 1000 peaks among the experiments to infer its PWMs; for the ranking method, see [8,22]). For each TF under study, we obtain up to 5 PWMs from an experiment or combined experiments. The coordinates of the PWM sequences in ChIP-seq peaks obtained in this study have been transformed to the latest version of the human genome (T2T CHM13v2.0/hs1) using the LiftOver tool (http://genome.ucsc.edu) [23].

**Inferring canonical and co-occurring motifs**

The canonical DNA binding motif of a TF refers to the sequence-specific DNA motif that is directly bound by the TF, while a co-occurring motif refers to a situation in which a second motif found in the ChIP-seq data of a TF (TF1) is similar to the canonical motif of a second TF (TF2) that belongs to another TF family. We use the following procedure to infer canonical and co-occurring motifs:

1. Among the inferred PWMs for a TF, the PWM that is most similar to known canonical PWMs in the same TF family that are available in public databases is chosen as the canonical PWM, if it exists. This PWM is not necessarily the top one. As will be discussed below, there are more complex situations than described here.

2. If none of the inferred PWMs for a TF is similar to any known canonical PWM in the same TF family and if the top PWM is not similar to any known PWM in all other TF families, then we consider it a "candidate canonical PWM.

3. If the top PWM is similar to a canonical PWM in another TF family, it is not the canonical PWM of the TF under study but represents the binding motif of a TF that belongs to a different TF family; that is, it is a co-occurring motif. Then, the next PWM is considered. This process is continued until either an inferred PWM is the canonical PWM of the TF or all inferred PWMs are exhausted.

In practice, there are more complex situations than the above. First, in the inferred PWMs from a data set, there can be more than one PWM that are similar to known canonical PWMs in the same TF family. In this case, all of the similar PWMs are considered canonical, but the one most similar to the known canonical PWMs in the same TF family is taken

as the representative. Second, a TF (e.g., CTCF) may have more than one representative canonical motif [24,25]. This situation can occur if a TF possesses multiple DBDs that some or all of them bind distinct motifs, or if a single long PWM is inferred as two or more PWMs. In the former situation, a canonical PWM is selected for each type of DBD. In the later situation, the multiple motifs are merged into one.

In this study, we used two PWM databases. The first one is CIS-BP (Weirauch et al. 2014), which includes a large number of PWMs inferred from SELEX data, which are *in vitro* binding data and so can well reflect the binding specificity of the TF. The second one is JASPAR (Castro-Mondragon et al. 2022), which includes manually curated canonical PWMs, so it has been commonly used to infer canonical PWMs.

## Co-binding vs. Tethered binding

Co-binding between transcription factors TF1 and TF2 is said to have occurred if both TF1 and TF2 binding motifs are found at two different sites on the same DNA fragment pulled down by the TF1 antibody while tethered binding is said to have occurred if only the TF2 binding motif but no TF1 binding motif is found on the DNA fragment. In co-binding, the two TF binding motifs both appear on the same ChIP-seq peak while in tethered binding no TF1 binding motif appears on the same ChIP-seq peak, although the ChIP-seq fragment was pulled down by the TF1 antibody (S1 Fig in S1 File).

For a TF, we first collect all of the PWMs inferred from each experiment. We then calculate the following motif counts in the ChIP-seq peaks across all available experiments for the TF:

X: Number of peaks containing only the canonical motif; all candidate canonical PWMs are now considered canonical PWMs.

Y: Number of peaks each containing at least one non-canonical (co-occurring) motif but no canonical motif.

Z: Number of peaks each containing both the canonical PWM and a non-canonical PWM.

The presence of a motif at a genomic location is determined using the PWM and FIMO with a p-value < 0.0001 [26] to screen the genome and a FIMO hit is retained only if it overlaps with a ChIP-seq peak. If the inferred PWMs for a TF do not include the canonical PWM but include only co-occurring PWMs in the studied experiments, we check if the canonical PWM is available in CIS-BP or JASPAR; if more than one is found, we choose the one with the highest information content (IC). Using the peaks thus inferred, we calculate the X, Y, and Z values for the TF under study. As described in the Results, we can use the X, Y and Z values to determine the relative frequencies of canonical binding, tethered binding, and co-binding in the ChIP-seq data for the TF under study.

To support our inference of co-binding between two TFs, we use the BioGRID database [27] to find evidence of interaction between the two TFs. According to the protein network of BioGRID, we check if two TFs bind each other directly or are bound by one or more co-factors, using the algorithm of shortest paths provided by the NetworkX tool [28]. All pairs of the TFs under study are searched, except for TFs for which canonical PWMs have not been found.

## Inferring core motifs of a TF family

We cluster the TF members from a TF family into groups based on PWM similarity (PCC > 0.80). The TF motifs in the same group are aligned using the ungapped Smith-Waterman alignment and used to obtain a consensus PWM for each group using STAMP [29]. The consensus motif is called a core motif of the TF family; a TF family may have more than one core motif. We only consider the TF families with at least 3 members that have canonical or candidate canonical PWMs; that is, a core motif has to be built by at least 3 PWMs.

## Positional distribution of TFBSs

We use the canonical and candidate canonical PWMs of TFs to infer TFBSs in the human genome. The procedure follows our previous study [8]. The key steps are: the PWM sequence of a TF in a ChIP-seq peak region is tested for conservation among 26 primates using phastCons30way [30], and it is said to be evolutionarily conserved if it is detected in more

than half of the species (score > 0.5). In the calculation of the positional distribution for all conserved TFBSs, two TFBSs are combined into one if they have an overlap of 90% or higher. The distance of a conserved TFBS (center position) with respect to the closest TSS is calculated using bedtools [31,32]. If a gene has more than one TSS, the distance of a TFBS to TSS is defined as the distance of the TFBS to the closest TSS in downstream/upstream. The TSS data were obtained from BioMart in Ensembl genome [33]. Additionally, in the calculation of statistical significance, random sites are generated using command "bedtools random -n N -l L -g Genome", where L is the length of the intervals to generate, N is the number of intervals to generate, and Genome is the length of each chromosome in the human genome.

### Annotation of TFBSs with respect to genomic features

We downloaded the candidate cis-Regulatory Elements (cCREs) data from SCREEN (version V3) and divided the cCREs into four biological regions: promoter-like signature (PLS), proximal enhancer-like signature (pELS), distal enhancer-like signature (dELS), and CTCF-bound sites [5]. The genomic regions of the TFBSs are annotated by annotatePeaks.pl in HOMER [34]. The fold changes of TFBSs of a TF in a biological region is calculated by $\log_2(N/N_{expected})$, where N is the number of TFBSs that located in the region and $N_{expected}$ is the expected number of TFBSs within the region.

## Results

### Newly discovered, revised and old TF binding motifs

We inferred the TF binding motifs (PWMs) from the ChIP-seq data released by ENCODE between July 2020 and January 2023 by the computational pipeline we put together in Yu et al. (2021) (S2 Fig in S1 File). This dataset included 552 TFs, of which 385 TFs were not included in Yu et al. (2021) (S2 Fig in S1 File). For a TF that has been studied in only one experiment, we used the top 500 ChIP-seq peaks to infer PWMs as we did in Yu et al. (2021), while for a TF that has been studied for two or more experiments, we used 1000 peaks instead because there were more peaks for selections (see Methods). For the 385 new TFs, we successfully obtained 441 PWMs for 196 TFs from 210 experiments; an experiment may provide up to 5 PWMs. For the remaining 189 (385–196) TFs, the data (202 experiments) did not meet our data quality control criteria, and so no PWMs were obtained for these TFs. From the 167 (552−385) TFs that have been included in Yu et al. (2021) but now have new additional experiments, we merged the new and old experiments and successfully obtained 1,423 PWMs for 119 TFs from 751 experiments. Combining the newly inferred and revised PWMs with our previous (old) PWMs for 235 TFs, we obtained 3153 PWMs for 550 (196 + 119 + 235) TFs. These data are given in Supplementary Table 1 in S2 File, and the database dbTFBS.v2 (https://dbtfbs.github.io/dbTFBS/?ver=codeBlue). Below, we will compare these PWMs with the known PWMs in public databases to infer the canonical PWMs for the 550 TFs under study.

### Identification of the canonical PWM of a TF

The canonical motif of a TF refers to the specific DNA sequence motif that is directly bound by the TF. Since multiple PWMs are usually inferred for a TF from an experiment or a set of experiments, we used a set of criteria to identify the canonical PWM for a TF (see Methods). Our procedure is briefly as follows. First, from the inferred PWMs, we identify those that are similar to known canonical PWMs in the same TF family in any public database; in this study, we used the two databases CIS-BP (Weirauch et al. 2014) and JASPAR (Castro-Mondragon et al. 2022). (In this study, two PWMs are said to be similar if their PCC ≥ 0.80, where the PCC is the Pearson correlation coefficient between the two PWMs.) As shown in Fig 1A, the canonical PWM may or may not be the top inferred PWM. If there are more than one inferred PWM that are qualified to be canonical PWMs, the PWM that is most similar to known PWMs is chosen as the representative canonical PWM (Fig 1B). Fig 1C gives an example that an inferred PWM may contain a sub-motif that is similar to the canonical PWM but contains an extra sequence that is dissimilar to the canonical PWM; such a PWM is not canonical.

Second, if none of the inferred PWMs is similar to the known canonical PWMs in the same TF family and if the top PWM is not similar to any known PWM in other TF families, then we consider the top PWM a "candidate canonical PWM".

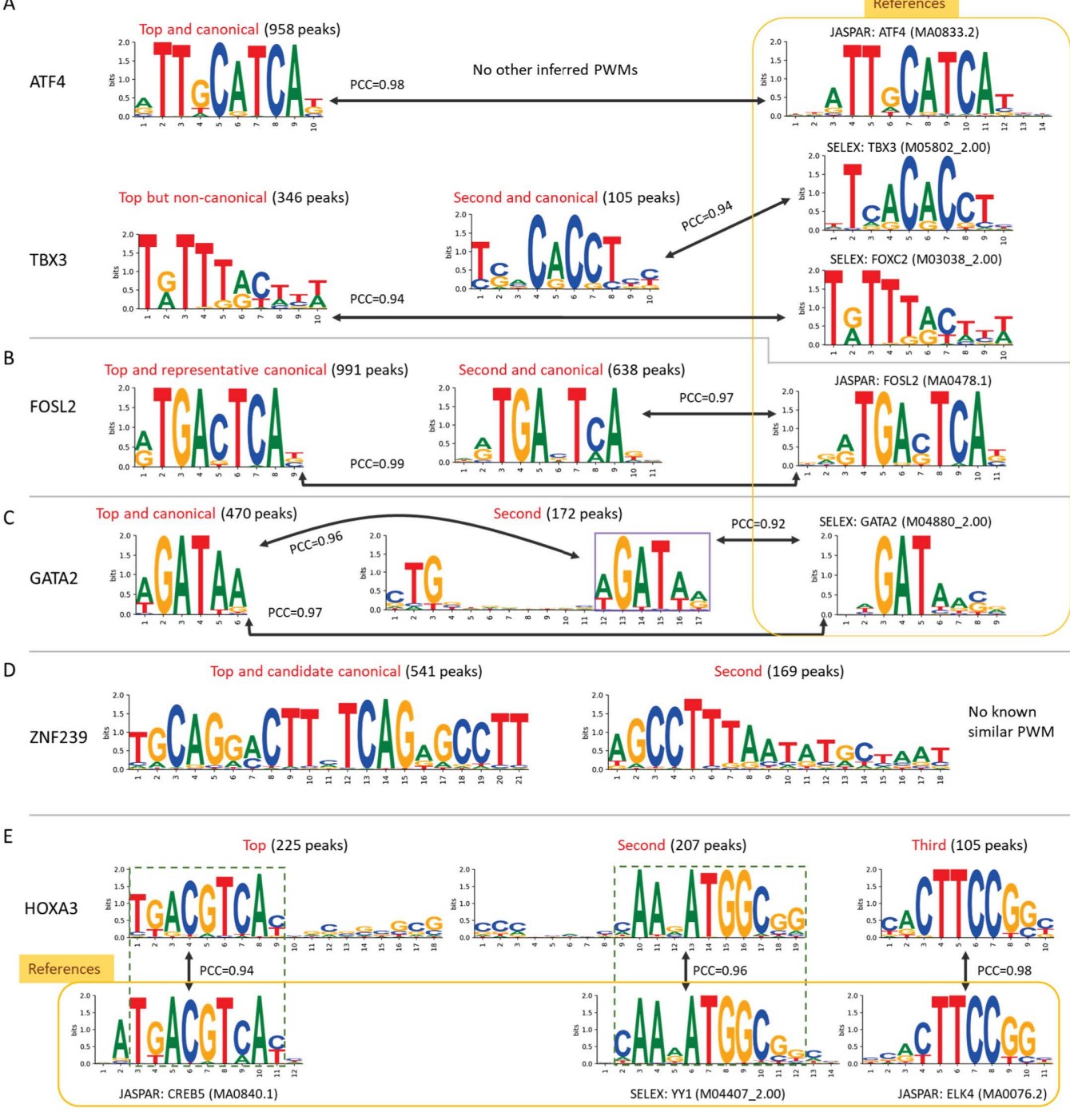

**Fig 1. Identification of the canonical PWM for a TF.** A. The identified canonical PWMs for TFs ATF4 and TBX3. For ATF4, there is only one inferred PWM and it is highly similar to the canonical PWM in JASPAR, so it is chosen as the canonical PWM. For TBX3, the inferred top PWM is highly similar to the SELEX PWM for TF FOXC2, so it is not the canonical PWM. In contrast, the inferred second PWM is highly similar to the canonical PWM in SELEX, so it is chosen as the canonical PWM. The Pearson correlation coefficient (PCC) values between two PWMs are indicated, and the number of peaks used to infer the PWMs shown in parentheses. B. Two canonical PWMs for FOSL2. Two inferred PWMs for FOSL2 are highly similar to the canonical PWM for FOSL2 in JASPAR, so both of them are canonical PWMs. As the first one of the two canonical PWMs is more similar to the PWM for FOSL2 in JASPAR, it is chosen as the representative canonical PWM. C. Canonical PWM for GATA2. The inferred top PWM is highly similar to the SELEX PWM for GATA2, so it is chosen as the canonical PWM. The second inferred PWM contains a sub-PWM that is highly similar to the top PWM but contains

an extra sequence, so it is not considered canonical. D. Inferred candidate canonical PWM for ZNF239. Two PWMs are inferred for the C2H2-ZF TF ZNF239. Neither PWM shows similarity to any known PWMs, so the top PWM is selected as the candidate canonical PWM. E. No inferred canonical PWM for HOXA3. For TF HOXA3, a homeodomain TF, three PWMs were inferred from the ChIP-seq data under study. None of them match any known canonical PWMs in the homeodomain TF family. Instead, they are highly similar to the canonical PWMs for CREB5 (bZIP), YY1 (C2H2-ZF), and ELK4 (Ets), respectively.

(We call it "candidate canonical" instead of "canonical" because it might be similar to an unknown motif of another TF not yet studied.) Fig 1D gives such an example.

Third, if the top PWM is similar to a canonical PWM in another TF family, it is not the canonical PWM but represents the binding motif of a TF that belongs to a different TF family. Then, the next PWM is considered and this process is continued until the canonical PWM is found or all inferred PWMs are exhausted. Fig 1E shows an example in which the three inferred PWMs are not similar to any known PWM in the same TF family but are similar to known canonical PWMs in other TF families. In this case, no canonical PWM for this TF is found in the ChIP-seq data under study.

Using the above procedure, we inferred the canonical PWMs for 279 TFs, in which 22 TFs have not been included in CIS-BP or JASPAR, and the "candidate canonical" PWMs for 221 TFs; thus, we have identified 243 (22+221) new canonical or candidate canonical PWMs (Supplementary Table 1 in S2 File). For the remaining 50 (550-279-221) TFs studied, only PWMs similar to those in other TF families were found; that is, no canonical or candidate canonical PWMs are found for these TFs. These PWMs will be discussed in "Co-occurring motifs". In the canonical and candidate canonical PWMs for 500 (279+221) TFs, 432 (86%) were the top PWMs, while only 68 (14%) TFs were not the top PWMs. All identified PWMs are given in Supplementary Table 1 in S2 File and the database dbTFBS.v2 (https://dbtfbs.github.io/dbTFBS/?ver=codeBlue).

### Core binding motifs of a TF family

In our previous study, we classified the canonical PWMs in a TF family into groups, using the criterion that two PWMs belong to the same group, if they are similar (i.e., PCC≧0.80); a group may contain only one TF. We then obtained the consensus PWM for each group that has three or more TFs and we called the consensus PWM a core motif of the TF family. Using the canonical PWMs inferred for the 550 TFs under study, we obtained PWMs for 46 TF families and the core motifs for 19 families. A TF family may have more than one core motif (S3 Fig in S1 File). For example, the 200 C2H2-ZF TFs under study were classified into 9 groups and we obtained 9 core motifs, which are highly diverse, indicating a highly heterogeneous TF family. The homeodomain and bHLH TFs studied have four and three core motifs, respectively. Some TF families such as Ets, E2F, GATA, and TEA each have only one core motif; that is, TFs in each of these families have very similar PWMs.

### Relative frequencies of canonical binding, tethered binding and co-binding

In a ChIP-seq DNA fragment pulled down by the antibody of the TF under study, three situations may occur (S1 Fig in S1 File). First, the DNA fragment contains only a canonical binding motif of the TF under study; sometimes it may contain two or more. Second, it contains only a binding motif of a TF belonging to a TF family different from that of the TF under study. Third, it contains both the canonical binding motif and a binding motif of a TF belonging to a TF family different from that of the TF under study. These three types of binding are known as "canonical binding", "tethered binding" and "co-binding", respectively. Now for a set of ChIP-seq data, let us compute the following three quantities: $X$ = number of peaks containing only the canonical motif (for simplicity, all candidate canonical PWMs are now considered canonical PWMs), $Y$ = number of peaks each containing no canonical motif but at least one non-canonical motif (i.e., tethered binding), and $Z$ = number of peaks each containing both the canonical motif and a non-canonical motif (i.e., co-binding). Then, we classify TFs into the following five groups: (1) $X > 0$, $Y = Z = 0$, i.e., canonical binding only, (2) $X > Y > Z$, i.e., canonical binding

frequency > tethered binding frequency > co-binding frequency, (3) X > Z > Y, i.e., canonical binding frequency > co-binding frequency > tethered binding frequency, (4) Y > X and Y > Z, i.e., tethered binding most frequent, and (5) Z > X and Z > Y, i.e., co-binding most frequent.

Among the 550 TFs under study, we found the following counts and relative frequencies for the five groups (Supplementary Table 2 in S2 File): (1) 274 (50%) TFs, (2) 50 (9%) TFs, (3) 17 (3%) TFs, (4) 189 (34%) TFs, and (5) 20 (4%) TFs. However, for the 550 TFs studied, there were 75 TFs that showed no canonical motif because no canonical PWMs were inferred from the ChIP-seq data we analyzed. We found the canonical PWMs for 42 of the 75 TFs from the SELEX data in CIS-BP (Weirauch et al. 2014) or from JASPAR (Castro-Mondragon et al. 2022) and for each of these TFs, we used FIMO with p < 0.0001 to screen the human genome and found that all 42 TFs had FIMO hits overlapping with the ChIP-seq peaks. These peaks were considered to contain the canonical motif, allowing us to examine the binding frequencies for these TFs. Among these 42 TFs, only 3 showed X > Y > Z; these TFs should be included in Group 2. The remaining 39 TFs all showed Y > X and Y > Z, so they belonged to Group 4. Using these new data, we revised the counts and relative frequencies for the five groups as (1) 274 (53%) TFs, (2) 53 (10%) TFs, (3) 17 (3%) TFs, (4) 153 (30%) TFs, and (5) 20 (4%) TFs. In Group 2 and Group 4, there remain no known canonical PWMs for 3 TFs and 39 TFs, respectively. When their canonical PWMs become known, we will make further corrections for the counts of the five groups. Note that in Groups 2 and 4, tethered binding is more frequent than co-binding while in Groups 3 and 5, co-binding is more frequent than tethered binding. Because Groups 2 and 4 are considerably larger than Groups 3 and 5, respectively, tethered binding occurs far more frequently than co-binding, consistent with previous findings [1,8,10–14].

## Co-binding TFs and co-factors

We found that 236 (99%) of the 238 TFs containing co-occurring motifs have experimental evidence, from BioGRID [27], of direct binding or binding mediated by one or more co-factors. The TFs containing co-occurring motifs were the TFs in all five groups except the group of canonical binding only, i.e., Group 1. Based on the experimental evidence of protein-protein interactions in Bio-GRID, we classified co-binding motifs into three groups (Fig 2A): (G1) the two TFs bind directly to two nearby motifs, (G2) the two TFs are bound together by one or more co-factors (N = 1 or >1) and then bind directly to two nearby motifs, and (G3) one TF binds two similar nearby DNA motifs. We estimated that the distances between two co-occurring sites (end-to-head in the same ChIP-seq peak) have a mean length of 42.6 bp for G1, 42.8 bp (N = 1) and 41.9 bp (N > 1) for G2, and 35.6 bp for G3 (Fig 2A). Note that the median distance for G3 (i.e., a TF binding to two motifs) is the shortest among the three groups. The co-occurring motifs (PWMs), their TFs, and co-factors are shown in Supplementary Table 3 in S2 File.

For the 35 TFs in G1, 14 TFs have an overlap with TFs in G2 and 12 have an overlap with TFs in both G2 and G3 (Fig 2B). For the overlapping TFs, most or all of them have different co-binding TFs, i.e., 11 TFs (11/14 = 78.6%) in both G1&G2 and all 12 TFs (12/12 = 100%) in G1&G2&G3, in which co-binding TFs were often found to be distinct among different tissues or cancer cell lines, suggesting that these TFs cooperate with different TFs. For the 207 TFs (G2) bound by one or more co-factors, only 18.4% (=38/207) of them have an overlap in G2 and G3 and a large portion of them (143/207 = 69.1%) need co-factor(s) to be able to bind DNA. For the 82 TFs that have nearby multiple sites (G3), 65.9% ((4 + 12 + 38)/82) of them can bind to other TFs directly or corporate with other TFs and co-factor(s), suggesting that these TFs may form regional modules to control expressions for downstream genes [27]. In addition, we find 225 TFs that show co-binding or tethered binding; 77 of them have been studied in more than one experiment (Fig 2C). The overlaps of co-binding TFs are summarized in S3 Table in S2 File.

## Positional distribution of TFBSs

We mapped the canonical and candidate canonical PWMs of TFs to the observed ChIP-seq peaks in the human genome to infer TFBSs (see Methods). The distribution of the inferred TFBSs shows a peak at the TSS and has substantial probabilities over a long distance from the TSS (Fig 3A). For each motif sequence found, we tested the evolutionary

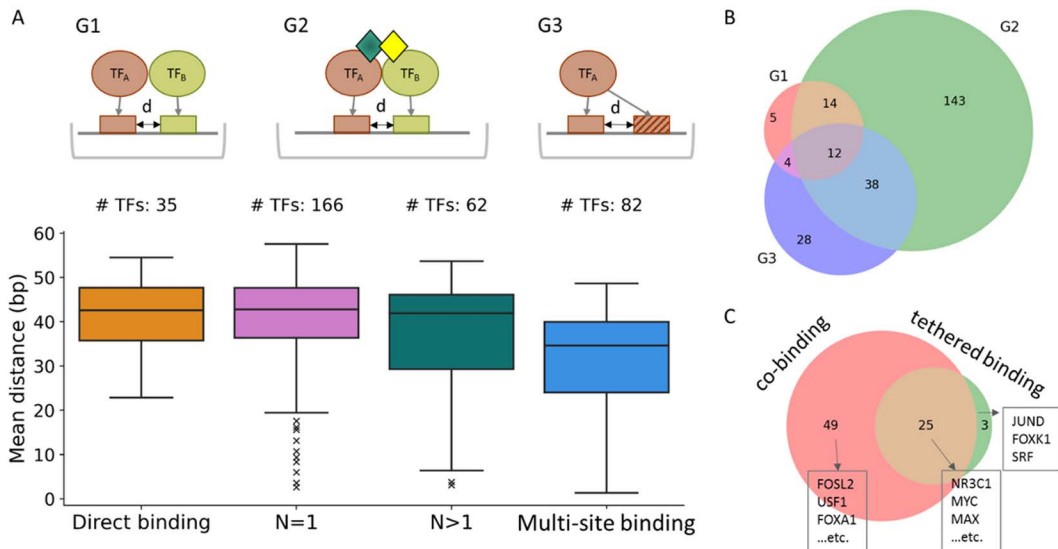

**Fig 2. Co-occurring TFBSs.** A. Distribution of the distance between two co-occurring TFBSs for three groups: (G1) Both TFs directly bind DNA. (G2) The two TFs are bound by one or more co-factors and then bind to two different motifs; there are two types: N = 1 (one co-factor) and N > 1 (more than one co-factor). (G3) One TF binds multiple sites in the same peak. The schema of binding types is shown in the upper panel and the number of TFs in each group is given at the top of the corresponding bar for the group. The "mean distance" in the Y-axis refers to the mean of the distances of nearby motifs in ChIP-seq peaks for the co-binding TFs or the multiple sites bound by a same TF. B. Venn diagram for the overlapping TFs between two groups or among three groups. C. Overlapping TFs between co-binding and tethered binding. The numbers of TFs are shown for those predicted to be co-binding and/or tethered binding; these TFs have been studied in more than one experiment. The three TFs studied with the largest number of experiments are listed for each case.

conservation of the sequence using the phastCons30way [30], requiring a conservation score > 0.5, i.e., it is found in over half of the 30 species, including 26 primates. Note that the TFBSs tend to occur within +/- 200 bp of the TSS and have a sharp peak at the TSS (Fig 3A). Interestingly, over 70% of the TFBSs are found in introns (42.6%) or intergenic regions (31.6%), whereas only 11.3% are in promoters (−1kb to +100 bp from a TSS). However, the intergenic regions and introns occupy 54.6% and 40.7% of the human genome respectively, so there is no TFBS enrichment in these regions, and TFBSs are actually enriched in promoters and 5' UTR (i.e., with $\log_2$ ratio = 3.3) (Figs 3C and 3D). The distribution of TFBSs is approximately symmetric with respect to the TSS of a gene. However, the density is slightly higher in the 3' side than in the 5' side of the TSS (Fig 3A), which might be due to better evolutionary conservation in the 3' side because the 3' side may have various functions. In another presentation, we scale the distance of a TFBS to the closest TSS by the gene length. Then, 72.7% of the TFBSs are in the gene body or its upstream one gene span (Fig 3B) and only 13.3% (/14.0%) of the TFBSs are in the upstream (/downstream) region beyond one gene span. In the intergenic region, TFBSs tend to be located around CTCF-bound sites (>3-fold more frequently than random) (Fig 3E). In gene body, the TFBSs tend to be in the first three exons (8.6%) and introns (49.7%) (Fig 3F).

To learn what kinds of TFBSs tend to be associated with CTCF-bound sites in intergenic regions, we rank the TFs under study by the association frequency of their TFBSs and the CTCF sites. In the SCREEN database [5], which provides a set of candidate cis-Regulatory Elements (cCREs) derived from ENCODE, there are a total of ~450,000 CTCF-bound sites, 46.2% of which are in introns, 34.3% are in intergenic regions, and 10.7% are in promoters. Among the CTCF-bound sites in intergenic regions (~154,000), 71.8% of them overlap with sites of distal enhancer-like signatures, suggesting that CTCF involves in the interactions between enhancers and promoters [35,36]. We find that 24.0% of CTCF sites contain one of our TFBSs within a region of 200 bases on both sides of the CTCF site. We calculate the percent

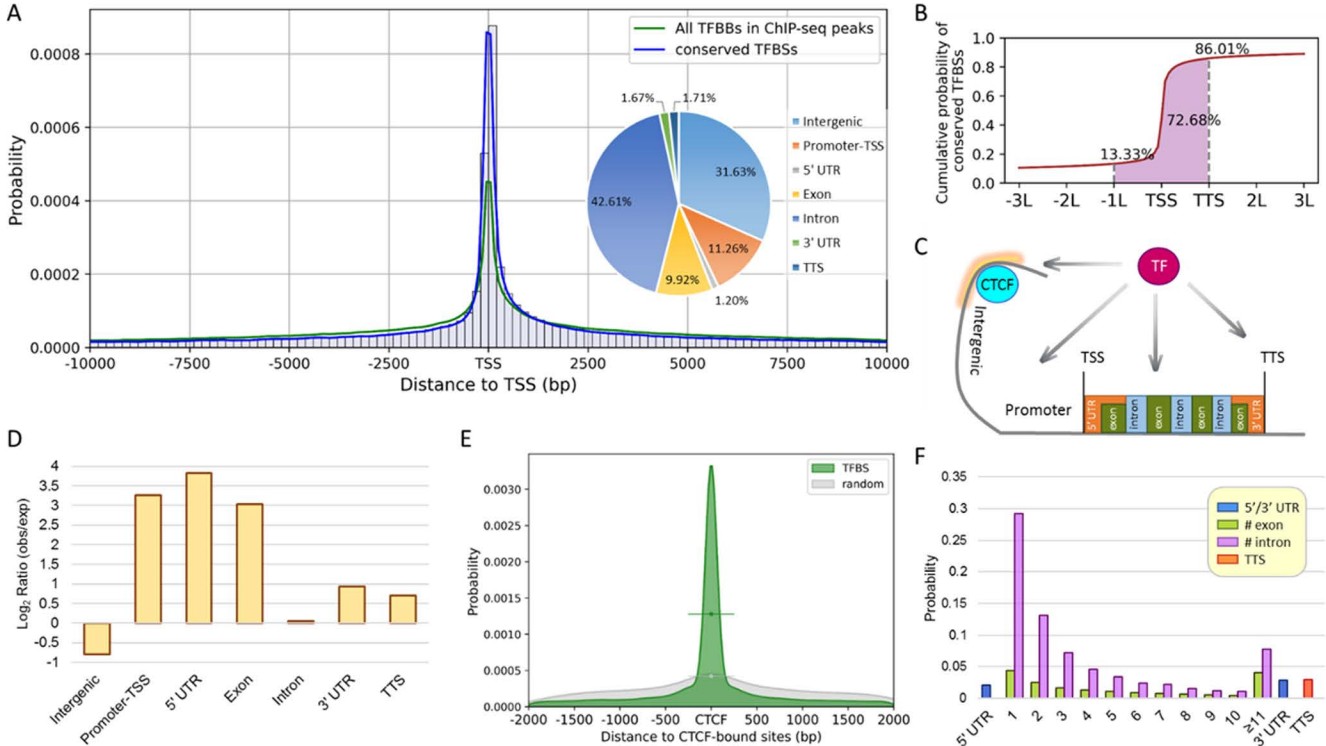

**Fig 3. Positional distribution of TFBSs in the human genome. A.** Positional distribution of TFBSs. The histogram shows the putative TFBSs of all available TFs located within 10kb of a transcription start site (TSS) and the total probability is normalized to one. A density curve (blue line) is fitted for the positions of the TFBSs conserved in primates by the method of kernel density estimation. For comparison, the green line shows the probability of all TFBSs in the ChIP-seq peaks. The probabilities of conserved TFBSs located in seven biological regions (including the locations outside of 10k bp) are shown in a pie chart. **B.** Cumulative probability distribution of conserved TFBSs in the scale of the gene length. The position of TFBSs (x-axis) is rescaled according to the gene length **(L)**, which is defined as the distance between the transcription start site (TSS) and the transcription termination site (TTS). The probabilities of TFBS occurrences are indicated for the regions in <-1L and in>TSS, and between TSS and TTS. Biological regions of TFBSs in the human genome include CTCF-bound sites, intergenic regions, promoters, transcription start sites (TSSs), 5' UTRs, exons, introns, 3' UTRs, and transcription termination sites (TTSs). **C.** The biological regions with their positions relative to the TSS indicated. These regions are referenced in the following figures. **D.** Enrichments of TFBSs in biological regions. The bar plot shows the $\log_2$ ratios of observed TFBSs to expected TFBSs in the seven regions. **E.** Distribution of the distances of TFBSs to nearest CTCT-bound sites. In the intergenic regions, the distance distribution of TFBSs (in green) and the distribution of random sites (in gray, see Methods) is shown with respect to the CTCF-bound sites. **F.** Preference probabilities of TFBSs in gene body. The gene body is divided into several regions, including 5' UTR, 1st to 10th and >10th exon/intron, 3' UTR and TTS.

abundance of a TF as the number of its TFBSs that are within 200 bp of a CTCF site divided by the total number of CTCF sites that have a nearby TFBS. The top 10 TFs are FOS (12.5%), JUND (11.0%), FOSL2 (9.6%), FOXA1 (9.6%), CEBPB (8.9%), RXRA (8.4%), MAZ (8.4%), FOXA2 (8.4%), IKZF1 (8.0%), and JUN (7.7%). Note that two different TFBSs may be close to the same CTCF site and two homologous TFs may bind the same TFBS, so the sum of the percentages is larger than 1 (not normalized to one). However, the top 10 TFs contribute to 37.6% of non-redundant CTCF sites that have a nearby TFBS. These 10 TFs belong to four TF families: bZIP (FOS, JUND, FOSL2, CEBPB, and JUN), forkhead (FOXA1 and FOXA2), nuclear receptor (RXRA), and C2H2 (MAZ and IKZF1) families. All the TFBSs that are close to CTCF binding sites are summarized in Supplementary Table 4 in S2 File.

In addition, we estimate the number of TFBSs in the promoters (−1kb to +100 bp of the TSS). Using the conserved TFBSs of the 492 TFs, a total of 16,965 promoters are found to have at least one TFBS, including 14,491 protein coding genes and 2,474 non-coding RNA genes. The average number of TFBSs per protein coding gene is 4.8 while the average

number of TFBSs per non-coding RNA (ncRNA) genes is 3.3. Note that the total numbers of protein coding genes and ncRNAs in human are 19,827 and 7,729, respectively, and so 73.1% of protein coding genes and 32.0% of ncRNAs are found to have at least one TFBS for the 492 human TFs under study. The total number of human TFs is 1,639, so the average number of TFBSs per gene (promoter) may be estimated as 4.8x1639/492 = ~16, a fairly large number! This estimate is an upper bound for the average number of TFBSs in the promoters because two TFs in the same TF family may have similar PWMs that may enable them to bind the same site.

Next, we consider whether different TF families have different distribution profiles. Since some TF families do not have enough TF members to provide sufficient data for computing the distribution, we select the seven largest TF families, each having >10 TFs studied, to compute familial distributions and put the remaining TFs into a group. Moreover, we focus on the distribution within the region from −2000 bp to +200 bp of the TSS. For the positional distribution of C2H2 TFBSs (Fig 4A), it has a median position (50% probability) at TSS and 10% of the TFBSs lie upstream of −818 bp. The C2H2-ZF TFBSs also have a relatively higher probability downstream of TSS, i.e., with 50% of the probability occurring downstream of the TSS and 90% of the probability occurring at position 98 bp (Fig 4A). In contrast, the TFBSs of Ets, bHLH, and Homeodomain TFs show higher probabilities around the TSS and their median positions are at −10 bp, −24 bp, and −50 bp, respectively (Fig 4B). For bZIP, nuclear receptor, and forkhead TFs, the TFBSs have wider spread distributions than the above four TF families (Fig 4C) and their median positions are at −43 bp, −48 bp, and −206 bp, respectively. Their distal 10% TFBSs lie at least 1.1 kb upstream of the TSS. For the remaining TF families, the distribution of the TFBSs shows the distal 10% TFBSs at −843 bp, the median position at −14 bp, and only 10% of the TFBSs lie 66 bp downstream of the TSS (Fig 4D). All the TF families under study except C2H2 have a median position of their TFBSs occurring upstream of the TSS.

## Association of TFBSs with biological features

Based on the spatial distribution of a TF, we calculated the locational preference of the TFBSs of a TF in terms of biological features to shed light on its potential function. The four biological features we consider are promoter-like signature (PLS), proximal enhancer-like signature (pELS), distal enhancer-like signature (dELS), and CTCF-binding sites; the data for these features are obtained from the SCREEN database [5]. For each region, the enrichment score of the TFBSs of a TF in a biological feature is measured by $\log_2$ fold change (FC). Among the 475 TFs under study, 457 have at least 10 TFBSs in one of the four biological regions. Statistically, most TFs (420 TFs) have FCs > 0 in PLS, among which 347 TFs have FCs > 1 in PLS (Fig 5A), suggesting that most TFs target the promoter sequence to regulate the expression of a gene. Moreover, the majority of TFs have TFBSs close to (<200 bp) CTCF-binding sites; that is, 51.6% TFs (=236 TFs) have FC > 0.5.

As the TFBSs of a TF show an uneven distribution in the four biological regions, we cluster the TFs into six groups based on the FC values using PCA (principal component analysis) for finer progressive patterns (Fig 5B). The TFs show a V-shaped profile on the PCA plot and have a strong correlation with the first PCA component (explained variance = 81.4%). For example, the TFs in groups G1 to G4 tend to bind TFBSs in PLS and near CTCF-binding sites and the strength of preferences decreases gradually as TFs move from G1 to G4 (Fig 5C). In addition, the FCs of TFs in dELS increase from G1 to G6 while the FCs of TFs in pELS show their highest enrichment with FC = 0.28 in G3 but becomes depleted toward either G1 or G6 (Fig 5C).

In pELS regions, TFs in the G3 group show a slight enrichment while the other groups show a slight (FC = −0.032), weak (FC = −0.089) or moderate (FC < −0.45) depletion, and in dELS regions, TFs in the G6 group show a slight enrichment (FC = 0.13), while TFs in G1, G2, and G3 show strong depletion (FC < −1.5) (Fig 5C). Note that the cut-off point of the distance between the definition of pELS and that of dELS is 2 kb from the TSS, so the skewness of the TFBS distribution toward pELS suggests that most TFs bind to enhancer-like regions within 2 kb of TSSs.

The TFs in the C2H2 family are found in all 6 groups but the majority are in G3 (Fig 5D). Indeed, some TF families show a biased distribution among the 6 groups. For example, seven of the eight E2F TFs under study are in G2, 10 of the

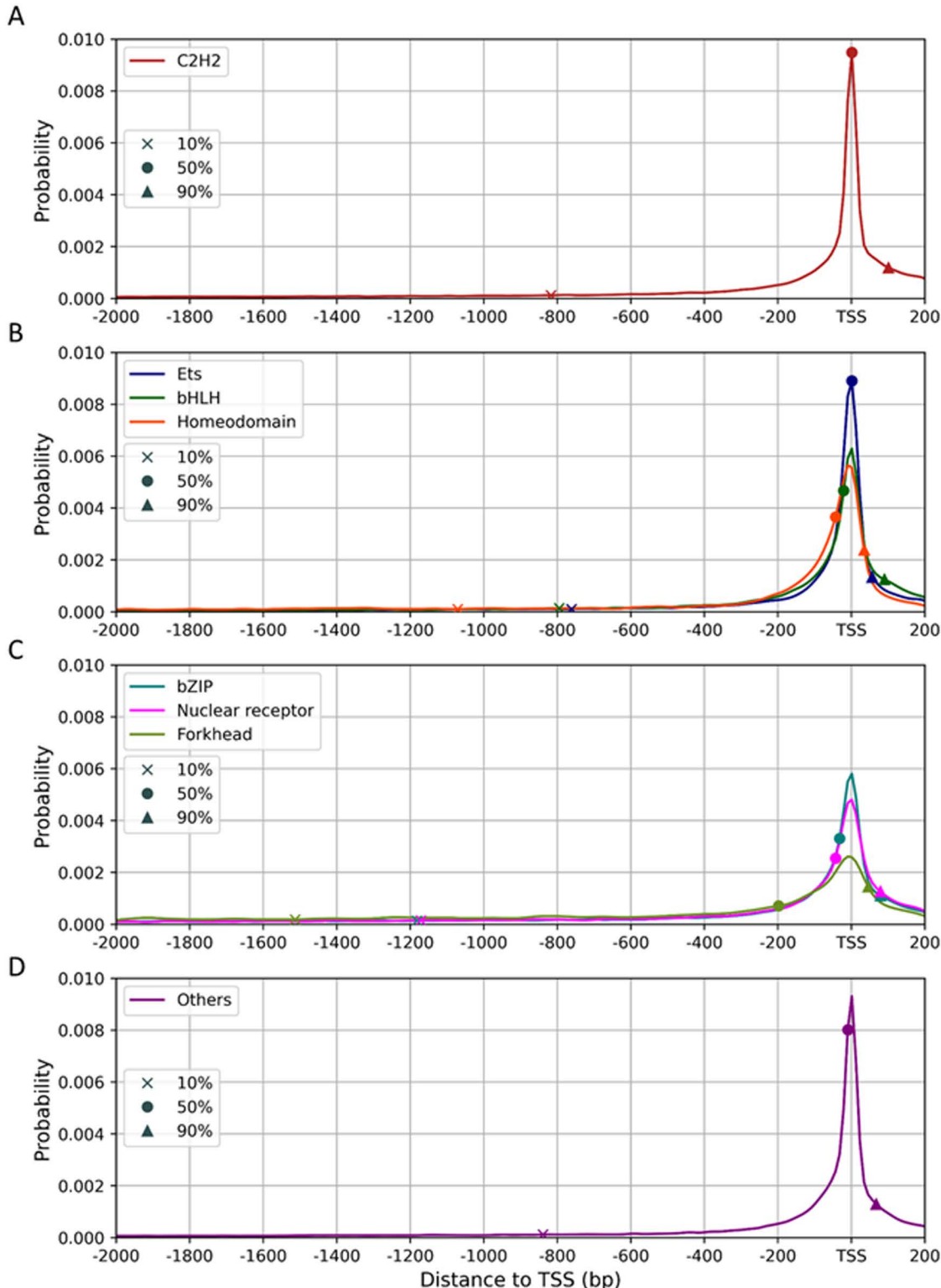

**Fig 4. Positional distributions of TFBSs in the region from −2000 bp to 200 bp of the TSS.** The seven largest TF families under study are classified into three groups, including (A) C2H2-ZF, (B) Ets, bHLH and homeodomain, and (C) bZIP, nuclear receptor and forkhead, and (D) the remaining TF families (see text). The distributions are shown in the region from −2000 bp to +200 bp of the TSS. In each distribution, the cumulative probability is marked at the 10%, 50% and 90% points by an X, a sloid circle and a solid triangle, respectively.

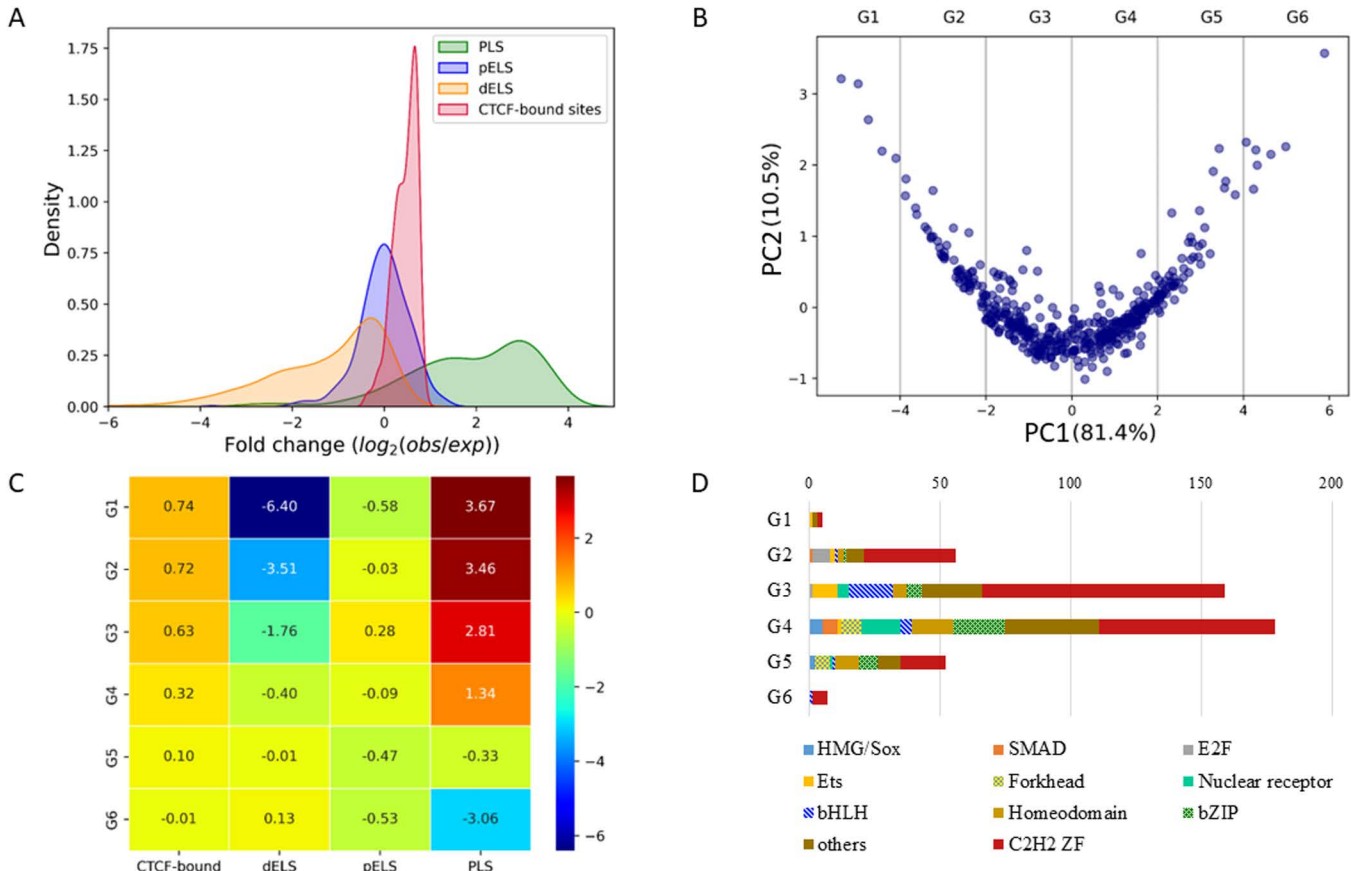

**Fig 5. Preferential distributions of TFBSs in genomic regions. A.** Fold-change (FC) distributions of TFBSs in each of the following four biological regions: promoter-like signature (PLS), proximal enhancer-like signature (pELS), distal enhancer-like signature (dELS), and CTCF-binding sites. The FC of a biological region is calculated by $\log_2$(no. of observed sites/no. of expected sites) for sites in the region of interest. **B.** Grouping of TFs by principal component analysis (PCA). A TF is represented by the FCs in the 4 regions and all TFs are mapped onto a 2-dimensional space using PCA. The explained variances of the two components are indicated in parentheses. Two principal components (PC1 and PC2) are each a linear combination of FCs in the 4 biological regions and PC1 contains most variance of the FCs (81.4%). We selected five cut-points (vertical lines) in PC1 to divide TFs into six groups (G1 to G6). **C.** Average FCs in the 6 defined groups. Each row shows the average FCs in 4 regions for each group. For example, TFs in G1 are enriched near CTCF-binding sites (FC = 0.74) and in PLS (3.7) but depleted in dELS (−6.4) and pELS (−0.58). **D.** TF family distributions in the 6 groups. The numbers of TFs in TF families are shown for each group. TF families are shown for ten largest families and the remaining TFs are assigned the "others" group.

14 Ets TFs are notably enriched in G3, and 17 of the 24 bHLH TFs show enrichment in G3, suggesting that TF families have a strong preference for biological regions, such as PLS and CTCF-binding regions. Note that the TF families of nuclear receptor, homeodomain, SMAD, and bZIP are enriched in G4.

## Discussion

In this study, we used the new ENCODE ChIP-seq data to infer the PWMs for 196 TFs and the combined new and old data to revise the PWMs for 119 TFs. By merging these PWMs with those of 235 TFs we inferred previously (Yu et al. 2021), we obtained the PWMs for a total of 550 TFs and identified the canonical PWMs for 500 TFs across 44 TF families. As ENCODE ChIP-seq data are of good quality and as 500 TFs are large enough for obtaining reliable statistics, we have

used these data to infer TF interactions, TF cofactors, the relative frequencies of co-binding and tethered binding, and the positional distribution of TFBSs in the human genome.

We found that the ranking method we used to select ChIP-seq peaks for inferring PWMs was effective for identifying the canonical PWM when multiple experiments were available. For instance, in the case of GATA3 (S4 Fig in S1 File), in the first experiment the first PWM was not canonical and in the third experiment both inferred PWMs were not canonical because both of them had a PCC < 0.80 with the SELEX PWM. In contrast, when all four experiments were combined using the ranking method, only one PWM was inferred and it closely matched the known canonical PWM of GATA3 (PCC = 0.96). Similarly, for PBX3 (S4 Fig in S1 File), while the canonical PWMs were mostly not the top PWM in individual experiments or no canonical PWM was inferred (the fourth experiment), applying the ranking method to the merged data identified the canonical PWM as the top motif, with a PCC = 0.90 to the known PWM of PBX3 in JASPAR. A reason for the improvement is as follows. In an experiment, a ChIP-seq peak may include either a single binding site bound by the TF under study or two binding sites bound by the TF under study and by a co-binding TF. When the ranking method is applied, the observed ChIP-seq peaks are prioritized based on their signal strength among multiple experiments. This prioritization reduces the likelihood of selecting sites bound by more than one TF, thus increasing the chance that the sites selected for inferring PWMs are sites bound by the TF under study. This also increases the chance that the top PWM is the canonical PWM. However, for understanding the TF DNA binding complexities, each experiment should be analyzed separately, as we have done for computing the relative frequencies of canonical binding, tethered binding and co-binding.

It was noted that most TFBSs are located in introns (42.6%) or intergenic regions (31.6%), while only ~11% are found in promoters. However, this distribution is due to the fact that the human genome is mainly occupied by introns and intergenic regions, and the density of TFBSs is in fact much higher in promoters than in introns and intergenic regions. Indeed, TFBSs show strong positional preference, as is evident from the following observations. The distribution of TFBSs in promoters has a sharp peak close to TSS and also in the promoters (from −2 kb to +200 bp of TSS) of most genes, over 90% of the TFBSs lie in the interval from −1 kb to +200 bp of the TSS and only <10% of them lie in the interval from −2 kb and −1 kb of the TSS. Note also that about half (49.6%) of the TFBSs in gene bodies are located in the first 3 introns. Thus, TFBSs seem to have a strong preference to be in or close to promoters. We also noted that TFBSs have a preference for enhancer regions.

A puzzling observation is that a large number (~208,000) of TFBSs are closely linked to CTCF binding sites. To understand the functional implications of this observation, we review the function of CTCF. When a CTCF protein and the cohesin complex together bind to a CTCF binding site to create a chromatin loop, the transcription of a gene occurs only if an enhancer and the promoter of the gene are both enclosed inside the loop (Holwerda and Laat 2013; Kim et al. 2015). Thus, a close linkage of a CTCF binding site to a TFBS increases the chance that the TFBS is inside the chromatin loop; that is, a close linkage would facilitate the transcriptional regulation of the gene associated with the TFBS. (Note that 'close linkage" means "less than 200 bp in distance".) This would particularly be true if the CTCF binding site is on or close to a promoter or an enhancer. Interestingly, we found that ~49,500 of the ~450,000 CTCF-binding sites in the human genome are in promoters. We also found that for the ~153,000 CTCF binding sites in intergenic regions, ~11,110 overlap with distal enhancer-like signatures. Finally, for the ~207,000 CTCF binding sites in introns, ~21,000 are within 1k bp of the promoter, so that when the CTCF and the cohesin complex create a chromatin loop, the promoter has a good chance to be inside the loop. It is interesting to note that the top 10 TFs (FOS, JUND, FOSL2, FOXA1, CEBPB, RXRA, MAZ, FOXA2, IKZF1 and JUN), which belong to the bZIP, forkhead, nuclear receptor and C2H2 TF families, contribute 37.6% of the total TFBSs that are closely linked to CTCF binding sites. The transcriptional regulation of the target genes of these 10 TFs might be facilitated by the close linkage to CTCF binding sites to their TFBSs. Supplementary Table 4 in S2 File lists the TFs that have TFBSs in proximity to CTCF sites.

Previous studies have found that a TF can co-bind DNA with another TF or can be tether-bound by another TF [8,11]. In this study, we analyzed a large number of TFBSs and found that a TF can have multiple co-binding TF partners and can be involved in tethered binding or co-binding in different tissues or cell lines. Another interesting finding is that two co-binding TFs often require two or more cofactors to form a protein complex, where some cofactors may not have any DBD, and we estimated the distances between two co-occurring motifs to be 35~43 bp. Also, we found that many TFs, mainly C2H2 ZF TFs, which tend to contain multiple and tandem DBDs, can bind not only canonical motifs but also other motifs at the same ChIP-seq peak. The canonical and secondary motifs at the same ChIP-seq peak may be bound by different DBDs or DBD clusters in a TF.

In conclusion, the present study revealed that interactions between TFs are more frequent and more complex than thought previously. The majority of the TFs studied can potentially form a homodimer or a heterodimer to bind a TFBS. Moreover, a TF can co-bind with one or more TFs to bind TFBSs at the same ChIP-seq peak and the co-binding may require one or more cofactors. In addition, tethered binding occurs far more frequently than co-binding. Future studies will likely reveal more interactions between TFs.

## Supporting information

**S1 File.   S1 Fig. Co-binding analysis.** Computing the counts of canonical binding, tethered binding and co-binding in a set of ChIP-seq data. **S2 Fig. Analysis flowchart.** Flow chart detailing the number of TFs with binding motif (PWM) inferred from ENCODE ChIP-seq data (July 2020 – January 2023). **S3 Fig. Core motifs.** Expanded identification across major TF families. **S4 Fig. Ranking method.** Identification of canonical PWM for a TF using the ranking approach.
(PDF)

**S2 File.    S1 Table. Canonical PWMs.** Summary of canonical PWMs for TFs and inferred PWMs from individual experiments. **S2 Table. Co-binding frequencies.** Frequencies of canonical binding, tethered binding and co-binding sites. **S3 Table. Co-occurring motifs.** Overview of identified motifs that frequently co-occur in binding regions. **S4 Table. TFBSs near CTCF-bound sites.** Summary of transcription factor binding sites within 200 bp of CTCF-bound regions.
(PDF)

## Acknowledgments

We thank Dr. John Wang for helpful suggestions. We are grateful to the National Center for High-performance Computing (NCHC), Taiwan, for providing computational and storage resources for this study.

## Author contributions

**Conceptualization:** Chun-Ping Yu, Wen-Hsiung Li.

**Data curation:** Chun-Ping Yu, Maloyjo Joyraj Bhattacharjee.

**Formal analysis:** Chun-Ping Yu, Zhi Thong Soh, Maloyjo Joyraj Bhattacharjee.

**Investigation:** Zhi Thong Soh.

**Methodology:** Chun-Ping Yu.

**Resources:** Wen-Hsiung Li.

**Supervision:** Wen-Hsiung Li.

**Writing – original draft:** Chun-Ping Yu.

**Writing – review & editing:** Zhi Thong Soh, Wen-Hsiung Li.

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
