## [Decision Letter · Decision Letter 0]

Dear Dr. Li,

Thank you for submitting your manuscript to PLOS ONE. We have now received reports from two reviewers. I am pleased to inform you that both reviewers found your study to be valuable and well-executed. They have raised only minor issues that should be addressed before we can proceed toward publication. Therefore, we invite you to submit a revised version of the manuscript that addresses the points raised during the review process.

We look forward to receiving your revised manuscript.

Kind regards,

Zhenguo Lin, PhD

Academic Editor

PLOS ONE

Journal Requirements:

National Science and Technology Council, Taiwan (NSTC 112-2311-B-001 -045)

We thank Dr. John Wang for helpful suggestions. This study was supported by National Science and Technology Council, Taiwan (NSTC 112-2311-B-001 -045). We are grateful to the National Center for Highperformance Computing (NCHC), Taiwan, for providing computational and storage resources for this study. Author contributions: The study was conceived by WHL and CPY and supervised by WHL. CPY, ZTS and MJB analyzed the data. All authors participated in writing the manuscript.

National Science and Technology Council, Taiwan (NSTC 112-2311-B-001 -045)

5. Please remove all personal information, ensure that the data shared are in accordance with participant consent, and re-upload a fully anonymized data set.

Reviewers' comments:

Reviewer's Responses to Questions

**Comments to the Author**

1. Is the manuscript technically sound, and do the data support the conclusions?

Reviewer #1: Yes

Reviewer #2: Yes

2. Has the statistical analysis been performed appropriately and rigorously?

Reviewer #1: Yes

Reviewer #2: Yes

3. Have the authors made all data underlying the findings in their manuscript fully available?

Reviewer #1: Yes

Reviewer #2: Yes

4. Is the manuscript presented in an intelligible fashion and written in standard English?

Reviewer #1: Yes

Reviewer #2: Yes

Reviewer #1: 1. In Abstract, the definition of ENCODE and CTCF should be provided when these two words first appear.

2. In Introduction, the definition of CTCF should be provided when it first appears.

3. In Inferring canonical and co-occurring motif sections, a “ is missing (“candidate canonical PWM).

4. How do the authors define TF families?

5. The definition of a co-occurring motif (in Inferring canonical and co-occurring motif sections) is strange. The definition that the authors used seems to have nothing to do with co-occurrence. Moreover, are co-occurring motifs (defined by the authors) biologically interesting?

6. The usage of co-occurring motifs and co-binding motifs is confusing. What are the differences between these two terms?

7. The authors classified 550 TFs into five groups. Please provide a browsable table in dbTFBS v2 so that users can easily access this information.

8. The authors clustered the TFs into six groups (Fig 5B). Please provide a browsable table in dbTFBS v2 so that users can easily access this information.

9. It would be great if the authors could provide a genome browser in dbTFBSv2 to show the genome locations and the target gene names of a TFBS of interest.

Reviewer #2: In this study, the authors used ENCODE ChIP-Seq data to infer position weight matrices (PWMs) for 500 transcription factors (TFs) across 44 TF families. Based on these data, they analyzed TF interactions, TF cofactors, the relative frequencies of co-binding and tethered binding, and the positional distribution of transcription factor binding sites (TFBSs) in the human genome. Their findings reveal that TF interactions are more frequent and complex than previously thought.

The manuscript is well-organized and clearly written. The findings provide novel insights into transcription regulation in human. However, a few issues should be addressed before publication:

1. For readers unfamiliar with transcription factors, the authors should include an annotation for CTCF(CCCTC-binding factor) when CTCF first appears.

2.In the PCA plot (Fig. 5B), the x- and y-axes should be relabeled as PC1 and PC2. The authors selected five cut-points in PC1 to divide TFs into six groups. The reason should be explained for clarity. Why not just divided into two groups in zero point?

**Do you want your identity to be public for this peer review?** For information about this choice, including consent withdrawal, please see our Privacy Policy

Reviewer #1: No

Reviewer #2: No

---

## [Author Response · Author response to Decision Letter 1]

7 Jul 2025

Our respnses to reviews are included in the uploaded fille.

---

## [Editor Report · Decision Letter 1]

Positional distribution of transcription factor binding sites in the human genome

PONE-D-25-30370R1

Dear Dr. Li,

We’re pleased to inform you that your manuscript has been judged scientifically suitable for publication and will be formally accepted for publication once it meets all outstanding technical requirements.

Kind regards,

Zhenguo Lin, PhD

Academic Editor

PLOS ONE
---

## [Editor Report · Acceptance letter]

PONE-D-25-30370R1

PLOS ONE

Dear Dr. Li,

I'm pleased to inform you that your manuscript has been deemed suitable for publication in PLOS ONE. Congratulations! Your manuscript is now being handed over to our production team.

Kind regards,

on behalf of

Dr. Zhenguo Lin

Academic Editor

PLOS ONE